# A Metabolomics Approach to Increasing Chinese Hamster Ovary (CHO) Cell Productivity

**DOI:** 10.3390/metabo11120823

**Published:** 2021-11-30

**Authors:** Grace Yao, Kathryn Aron, Michael Borys, Zhengjian Li, Girish Pendse, Kyongbum Lee

**Affiliations:** 1Bristol Myers Squibb, Biologics Development, Devens, MA 01434, USA; grace.yao@tufts.edu (G.Y.); karon@obsidiantx.com (K.A.); michael.borys@bms.com (M.B.); zli@horizontherapeutics.com (Z.L.); girish.pendse@bms.com (G.P.); 2Department of Chemical and Biological Engineering, Tufts University, Medford, MA 02155, USA

**Keywords:** Chinese hamster ovary (CHO) cells, cell-specific productivity, metabolomics, citrate, aspartate

## Abstract

Much progress has been made in improving the viable cell density of bioreactor cultures in monoclonal antibody production from Chinese hamster ovary (CHO) cells; however, specific productivity (qP) has not been increased to the same degree. In this work, we analyzed a library of 24 antibody-expressing CHO cell clones to identify metabolites that positively associate with qP and could be used for clone selection or medium supplementation. An initial library of 12 clones, each producing one of two antibodies, was analyzed using untargeted LC-MS experiments. Metabolic model-based annotation followed by correlation analysis detected 73 metabolites that significantly correlated with growth, qP, or both. Of these, metabolites in the alanine, aspartate, and glutamate metabolism pathway, and the TCA cycle showed the strongest association with qP. To evaluate whether these metabolites could be used as indicators to identify clones with potential for high productivity, we performed targeted LC-MS experiments on a second library of 12 clones expressing a third antibody. These experiments found that aspartate and cystine were positively correlated with qP, confirming the results from untargeted analysis. To investigate whether qP correlated metabolites reflected endogenous metabolic activity beneficial for productivity, several of these metabolites were tested as medium additives during cell culture. Medium supplementation with citrate improved qP by up to 490% and more than doubled the titer. Together, these studies demonstrate the potential for using metabolomics to discover novel metabolite additives that yield higher volumetric productivity in biologics production processes.

## 1. Introduction

Chinese hamster ovary (CHO) cells are widely used for manufacturing of biopharmaceutical proteins, especially monoclonal antibodies (mAbs). Maximizing upstream productivity allows for greater facility flexibility, smaller footprints, and lower costs for patients. Over recent decades, CHO cell culture processes have gradually improved in volumetric productivity to reach recombinant protein titers on the order of 10 g/L in fed-batch production [1,2,3]. Much of the titer increases seen in the past few years have been due to increased viable cell density (VCD) and sustained viability [4,5]. In comparison, cell-specific productivity (qP) has improved to only a limited extent [6]. As the biomanufacturing industry implements process intensification technologies such as perfusion cell culture in integrated continuous bioprocesses, CHO cell culture processes may soon approach the upper limits of achievable VCDs. This would constrain any further improvements to volumetric productivity. Therefore, there is an urgent need to identify efficient strategies for developing high-producing clones and optimizing media or process conditions not only for VCD but also qP.

To investigate cellular mechanisms leading to different productivity phenotypes in CHO cells, many omics approaches have been used, including transcriptomics and proteomics [7,8,9,10]. Although challenges persist in metabolite identification, untargeted metabolomics has been increasingly adopted for characterizing the metabolic profiles of industrially relevant CHO cells cultured in bioreactors [11,12]. Recent studies have used metabolomics experiments to identify metabolites associated with apoptosis, hypoxia and ROS generation, or growth inhibition [13,14,15]. Only a few studies have examined cellular productivity [9]. Notably, Chong et al. analyzed intracellular metabolites from several CHO cell clones showing high or low productivity and found that metabolites related to redox status regulation were elevated in high producers [16]. Dean and Reddy compared metabolic fluxes between a low- and a high-productivity cell line using isotope labeling experiments, identifying differences in glycolysis, the TCA cycle, and lactate metabolism; Templeton et al. found similar differences among multiple low- and high-productivity cell lines [17,18]. Recently, Huang et al. used targeted metabolomics data in conjunction with a genome scale modeling approach to optimize media for enhanced qP [19]. Qian et al. also used targeted metabolomics to connect a decrease in qP over time to elevated lipid oxidation in unstable cell lines [20].

These studies have provided important clues linking CHO cell metabolism to qP. However, it remains an open question how metabolic differences drive qP differences and how cellular metabolism could be manipulated to improve qP. In this work, we analyze a library of clones with varying qP producing different mAbs to identify endogenous (CHO cell derived) metabolites that are positively associated with qP and could be used as productivity indicators or medium additives to improve clone selection and cell culture process development. Extracellular samples generated from an initial library of 12 clones with a wide range of qP were used for an untargeted LC-MS metabolomics study. Metabolic model guided annotation and correlation analysis identified a panel of metabolites significantly and specifically associated with qP. Targeted metabolomics experiments using a different set of 12 mAb producing clones confirmed the positive correlations between qP and amino acids identified by the untargeted study. Furthermore, supplementing the culture medium with a qP-associated central carbon metabolite, citrate, significantly improved qP and final titer.

## 2. Results

### 2.1. Untargeted Metabolomics Identified Endogenous CHO Cell Metabolites That Correlate with qP but Not Growth

A set of 12 clones, six expressing mAb A and six expressing mAb B at industrially relevant titers on the g/L scale, was selected to represent a wide range of growth (Figure 1A), titer (Figure 1B) and productivity profiles (Figure 1C). All 12 clones were cultured in identically operated 5 L bioreactors using the same fed-batch platform process. Detailed qP profiles at different time points are shown in Appendix A. For all clones, VCD peaked between days 6 and 8, with peak VCD ranging from about 18 to 39 × 10^6^ cells/mL. Titer and qP were both the highest for clone A-5. Interestingly, although often higher growth can lead to higher volumetric productivity, clone B-6 displayed the lowest titer and the highest growth, indicating that growth and volumetric titer are not always positively correlated. Across the 12 clones, we detected a significant negative correlation was found between peak VCD and qP (Appendix A).

To determine the clones’ metabolite profiles, supernatant samples were collected on days 4 and 7, corresponding to mid-exponential and early stationary phases of the bioreactor cultures. Untargeted LC-MS analysis on the supernatant samples detected 4541 features between the four combinations of LC methods (HILIC or RP, see Materials and Methods) and MS modes (ESI positive or negative), including adducts and features that were detected by multiple methods. Analysis of the features’ accurate mass (*m*/*z*) and MS/MS spectra using a semi-automated annotation tool [21] mapped 167 of these features to CHO cell metabolites. A subset of these metabolites was selected for testing against high purity chemical standards to verify the annotations (Appendix A).

Metabolite abundances, represented by extracted ion chromatogram peak areas (AUCs), were normalized to integral VCD to control for the clones’ growth rate differences (Figure 2A). Out of the 167 annotated features, 70 features were duplicates detected in more than one analysis method and were eliminated from the correlation analysis as described in the methods section. The remaining 97 annotated and unique features were tested for significant correlations with cell growth or protein production. Pearson and Spearman rank correlation coefficients calculated between each metabolite and average qP or peak VCD gave similar results. Pearson coefficients are shown in Figure 2B and Appendix A. There were no significant correlations for the mid-exponential phase (day 4) samples. Early stationary phase (day 7) measurements (normalized AUCs) were significantly correlated with qP, growth, or both for 73 of the 97 features. More than half of the 73 features reached at least level 2 identification (Appendix A) based on minimum reporting standards of the Metabolomics Standards Initiative (MSI) [22]. Out of the 73 putatively identified and significantly correlated metabolites, about 25% are part of the basal or feed medium, indicating that most of the correlated metabolites result from endogenous metabolism of CHO cells. A delta analysis between day 4 and day 7 iVCD-normalized AUCs was performed to examine whether any cell-specific rates were correlated with qP or growth. In this analysis, 11 metabolites showed a significant positive correlation with qP, and 2 of these showed an additional significant negative correlation with growth. All 11 metabolites were also significant in the day 7 correlation analysis. Notably, none of them were originally part of the feed or basal medium, indicating that their accumulation was not due to reduced net uptake.

In general, metabolites significantly correlating with qP had positive associations (higher extracellular metabolite levels correlated with higher qP), while those significantly correlating with growth had negative associations (Figure 2B). Only three metabolites had negative correlations with qP, while one metabolite had a positive correlation with growth. None of these correlations were significant. This result is consistent with several earlier studies that have shown negative associations between growth and endogenous (CHO cell produced) metabolites accumulating in the culture medium [23]. Positive associations between qP and accumulated metabolites could reflect an inverse correlation of qP with growth, possibly due to the metabolic burden of recombinant protein biosynthesis [24]. However, it is important to note that there are some metabolites, e.g., aspartate, that showed no correlation with growth but still positively correlated with qP.

Pathway impact and enrichment analysis on the set of metabolites correlated with qP, including those also correlated with growth, using Metaboanalyst showed that the two pathways with both high pathway impact and enrichment of the metabolites that have significant correlations with qP were the TCA cycle and alanine, aspartate, and glutamate metabolism had (Figure 2C) [25]. Meanwhile, enriched pathways for metabolites that only correlated with growth were aminoacyl-tRNA biosynthesis and tyrosine metabolism (data not shown). The latter is consistent with a recent study reporting a beneficial impact of tyrosine on cell growth [26].

### 2.2. Targeted Analysis Confirmed Aspartate and Cystine as qP Specific Metabolite Indicators

A fed-batch study with a new set of 12 clones producing a third mAb (molecule C) was performed in Ambr 15 bioreactors to investigate if significant metabolites identified by the untargeted analysis might serve as useful metabolic indicators of high-qP cell lines. Like the clones producing molecules A and B, these clones exhibited a wide range of growth and productivity profiles (Appendix A).

Supernatant samples from day 7 were analyzed using targeted LC-MS experiments for 34 metabolites that were significantly and positively associated with peak VCD and/or qP in the untargeted analysis. The selection of metabolites was based on commercial availability of standards. The day 7 time point was selected to allow discovery of early indicators of qP and to avoid potentially confounding influences from nutrient starvation that may occur towards the end of a bioreactor run. The targeted analysis again found that aspartate was positively correlated with qP, confirming the association from the untargeted experiment (Figure 3A,C). In addition, cystine was tested, as it had a significant negative correlation with growth and a positive correlation with qP, although the latter correlation did not meet the significance threshold of FDR-controlled *p*-value less than 0.05. In the targeted analysis, cystine was found to be significantly and positively correlated with qP (Figure 3B,D). Aspartate and cystine were not significantly correlated with growth in the targeted analysis (Appendix A). This indicates that a positive correlation between these two metabolites and qP does not merely reflect the negative correlation between qP and growth.

### 2.3. Medium Supplementation with qP Correlated Metabolite Improved qP and Titer

We next investigated whether the metabolites that significantly correlated with qP could be used to improve qP and antibody titer. To this end, an addback study with a D-optimal design was performed with selected metabolites from the TCA cycle and from alanine, aspartate, and glutamate metabolism based on the observation that these two pathways were enriched in the set of metabolites positively correlated with qP (Figure 2C) [27]. The selected metabolites were aspartate, glutamate, 4-aminobutanoate (GABA), and citrate. The lowest- and highest-qP clones expressing mAb A and mAb B were cultured in 50 mL conical tubes using a scaled down version of the fed-batch platform process. The selected metabolites were added to the cultures on day 3 at varying concentrations based on a D-optimal design (Appendix A). By day 3, the cells have entered exponential growth, but have not begun antibody production. Day 3 was also early enough to detect potentially negative effects of metabolite supplementation on peak VCD as a proxy for overall growth.

Of the four metabolites tested, only citrate had a clear dose-dependent effect on qP (Figure 4 and Appendix A). The low and medium concentrations of citrate improved qP in three of the four clones, but negatively impacted growth. The high concentration was detrimental to both growth and qP. In the low-qP clone A-2, the low dose of citrate had minimal impact on growth, while improving qP two-fold, resulting in a higher overall titer compared to the control without citrate supplementation.

To determine if citrate addition could improve the qP of other clones and confirm that this effect also occurs under bioreactor conditions, a similar addback experiment was performed with a narrower concentration range using Ambr 250 cultures. Three of the clones expressed molecule A, and three expressed molecule B. In all clones, citrate supplemental reduced peak VCD in a dose-dependent manner (Appendix A). This effect was modest, however, with the highest concentration reducing peak VCDs by 10–35% compared to the controls. In five out of the six clones, citrate addition improved qP in a dose-dependent manner (Figure 5A). In four clones, the qP improvement outweighed the growth reduction, leading to an increase in overall volumetric titer of 15–107% over the controls without citrate supplementation (Figure 5B). The qP and titer improvements were observed for both mAb A and mAb B producing clones. Dose-dependent changes in metabolite profiles were observed for the five clones showing a positive qP response to citrate addition, including increased accumulation of glutamine, glutamate, and lactate, and reduced accumulation of ammonium (Appendix A). Interestingly, clone A-1, which did not show a positive qP response to citrate addition, also did not show these metabolic responses.

## 3. Discussion

In this work, a metabolomics approach was used to identify CHO cell metabolites that are associated with productivity, cell growth, or both, across multiple clones producing different mAbs. These metabolites were also tested for their potential utility in cell culture process development either as a qP indicator or medium additive. These studies identified aspartic acid and cystine as potential clone-independent indicators of qP. The correlation between these metabolites and qP observed in one set of clones producing two different mAbs was validated in a distinct, second set of clones expressing a third mAb. Furthermore, this work showed citrate can function as a culture additive benefiting both qP and volumetric titer in multiple clones under production process conditions.

### 3.1. Tradeoff between Cell Growth and mAb Production

We observed that significant correlations between metabolites accumulating in the fed-batch cultures and growth were almost always negative, whereas correlations with qP were positive. The negative association between growth and metabolite levels observed in the present study is consistent with several previous studies that reported on growth-inhibitory effects of metabolic byproducts, including many of the TCA cycle metabolites, amino acids, and their derivatives reviewed by Pereira et al. [16,23]. Well-known examples of growth-inhibitory byproducts include ammonium and lactate, the accumulation of which has been consistently reported to have a negative impact on cell viability and growth [28,29,30,31]. Recently, it has been shown that reducing the accumulation of branched chain amino acid (BCAA) catabolites, either by decreasing concentrations of the amino acid precursor or by metabolic engineering of the cell line to limit BCAA transamination, can lead to better growth [14,32]. Alden et al. found that a tryptophan metabolite, 5-hydroxyindole acetaldehyde, negatively correlated with growth, and that tryptophan addition led to growth inhibition [33]. Although efforts in understanding growth impacts of byproducts can be used to improve productivity, higher growth may not always lead to increased titers, as qP may be negatively affected by process modifications that favor growth over protein production.

Compared to cell growth, very little has been reported regarding the impact of accumulating metabolites on qP. Unlike cell proliferation, production of a mAb is not native to CHO cells and could place metabolic burdens on the cell that are not subject to endogenous regulatory mechanisms, i.e., the cell is not naturally programmed to achieve balanced mAb production. A reduction in qP could reflect a depletion of biosynthetic precursors or compromised cell viability. In principle, a depletion of carbon, nitrogen or energy resources needed for heterologous protein production could be mitigated by media supplementation or by engineering increased flux through pathways that supply the precursors. However, a reduction in qP due to compromised cell viability could be more complex to address. Collectively, the trends from the fed-batch studies referenced in the previous paragraph suggest declining viability in fed-batch systems is likely due to accumulation of harmful metabolites, rather than nutrient depletion. This is also supported by studies showing that cell density and productivity, as well as high viability, can be maintained for long periods of time in perfusion systems which continuously remove toxic byproducts [34,35]. Conversely, an increase in qP could indicate that more resources are directed towards protein production instead of cell growth and maintenance. Cell growth and heterologous protein production could require different balances of nutrients, and if the culture is operated the same way during both exponential and stationary phases of a fed-batch process, accumulation of metabolites may be seen.

In the present study, we show that almost all significant correlations between qP and metabolites detected in the culture medium are positive, in contrast to correlations between peak VCD and the metabolites. If a metabolite is accumulating in a culture, the cells are consuming less of it (if it is in the feed or basal media), making an excess of it, or there is a bottleneck in the metabolite’s use. This accumulation is more likely to happen when cells enter stationary phase, growth slows, and a larger share of metabolic resources (biosynthetic precursors and energy currency metabolites) is consumed for producing the non-native molecule [36]. Before this point, accumulation of metabolites may not differ significantly between low-qP and high-qP cultures. This is consistent with the results of our untargeted analysis on mid-exponential phase (day 4) samples. After the stationary phase begins (day 7), we see that the metabolic profile of cells that grew to a high cell density is different than the metabolic profile of cells that exhibit high qP.

The best-growing cells likely have a flux distribution optimized for growth with minimal overflow of central carbon metabolites. Meanwhile, accumulation of central carbon pathway intermediates may reflect a suboptimal flux distribution for growth but more capacity for a non-natural objective such as producing recombinant protein. This tradeoff could explain why only negative correlations were seen with growth and positive correlations with qP. Additionally, cell culture processes are generally optimized for titer rather than growth, with many processes even limiting proliferation in favor of producing antibody [37,38]. The platform process used in these studies was indeed previously optimized for antibody expression in the parental cell line, which could be another reason most correlations with qP were positive.

### 3.2. Aspartic Acid and Cystine

Metabolites that correlate with qP across different clones independent of the clones’ mAbs could be useful early indicators for selecting high-qP clones. Ideally, such indicators specifically predict qP and are independent of growth. In this work, day 7 aspartic acid and cystine levels in the culture medium were identified as potential indicators that had statistically significant correlations with qP of 24 clones expressing three different molecules. Furthermore, both metabolites showed no correlation with growth in the targeted analysis. Extracellular metabolites are easily measured indicators that could be used to assess the culture at an early stage of protein production (e.g., day 7 of a 14-day process) to reduce timelines during multiple rounds of clone selection. The indicators could also reveal clones with high-qP potential that may not perform well in initial clone selection due to unoptimized platform culture conditions such as starvation towards the end of a process. Although both aspartic acid and cysteine were present in the basal and feed media used in the study, there were differences in the consumption rates of these amino acids between low-qP and high-qP clones.

A previous study reported that aspartate is produced by mammalian cells when the cells are grown in high glucose/glutamine conditions, whereas the amino acid is consumed by the cells in low glucose/glutamine conditions [39]. In the presence of excess glucose, as was the case in the present work, aspartate production may have occurred as a byproduct of elevated glucose metabolism in the higher qP clones. Alternatively, the positive correlation between qP and aspartate could reflect the amino acid’s involvement in the malate-aspartate shuttle [40]. Overexpression of an aspartate/glutamate carrier in the malate-aspartate shuttle has been shown to increase ATP production in CHO cells [41]. A high concentration of ATP could deactivate adenosine monophosphate kinase (AMPK) and modulate mTOR signaling, which has been found to promote qP [42,43]. The positive correlation of aspartate with qP could also be linked to increased oxidative phosphorylation, a phenotype associated with higher productivity phases and clones [18,44].

Cystine was measured in the present study instead of free cysteine, which readily dimerizes to cystine in the presence of oxygen. The accumulation of cystine in high-qP clones could indicate a more oxidized cell culture environment. Previous studies have reported that having sufficient cysteine in the medium supports high productivity [45]. Cysteine is an important substrate for synthesis of antioxidants, including taurine and glutathione (GSH), which support high rates of oxidative phosphorylation associated with high-qP cells. A recent study found that a depletion of cysteine in CHO cell culture medium negatively impacted VCD, viability, and qP, and attributed these observations to redox imbalance, endoplasmic reticulum stress, and mitochondrial dysfunction [46]. The accumulation we observed in high-qP cells could reflect the ability of these cells to supplement feed cysteine with endogenous production. We did not observe a significant trend with methionine, the other sulfur-containing amino acid, but the cells could be sourcing the sulfur by turning over proteins. Further studies, e.g., using isotopic tracers, are warranted to better understand the source of cystine in the medium, the mechanisms behind metabolite-qP correlations and to evaluate more broadly the use of the metabolites as biomarkers of qP.

### 3.3. Citrate Addition Improves qP and Titer

Addition of citrate to the medium on day 3 increased qP for multiple clones in a dose-dependent manner. In the untargeted experiment (Figure 2), citrate correlated positively with qP but negatively with growth. Previous studies have suggested that the accumulation of TCA cycle intermediates in the media may signal a bottleneck that is connected to growth limitation [47,48,49]. In the present study, multiple clones showed a benefit to qP with citrate addition that overcame any impact on growth. One possible explanation for our finding is that metabolism in the clones benefiting from citrate addition has not yet reached a bottleneck, and the added citrate increased the amount of substrate available for the TCA cycle. Zhang et al. postulated this hypothesis in an earlier study which found that feeding various TCA cycle intermediates led to titer increases [50]. Another study reported that feeding TCA cycle intermediates, including citrate, to CHO cells during stationary phase resulted in lower ammonium accumulation and higher glutamate and glutamine concentrations in the culture medium [51], consistent with findings in this study (Appendix A). These observations could be explained by additional conversion of citrate to α-ketoglutarate (aKG), which would reduce transamination of glutamate and production of ammonium, while increasing flux of glutamate to glutamine. Entry of additional citrate into the TCA cycle could also reduce the demand for cytosolic pyruvate, leaving more pyruvate available for reduction to lactate. This scenario is consistent with the lactate profiles observed in this work. However, current literature suggests that elevated pyruvate flux towards lactate indicates a less efficient metabolic state for antibody production [18,33]. Further investigation into intracellular carbon fluxes may be able to resolve this difference.

An alternative explanation for citrate’s effect on qP is that it promotes iron chelation. An earlier study reported that adding sodium citrate in combination with ferric sulfate increased qP [52]. A future study comparing citrate with the effects of other iron-chelating compounds while examining ATP production in all phases of the cell culture could shed light on the mechanism by which productivity increased. Quality attributes including glycan distribution and charge variants should also be considered in future studies.

The titer improvement seen in this study from medium supplementation of a metabolite identified through untargeted metabolomics on a library of clones followed by correlation analysis shows that this approach has the potential to discover early indicators of qP and beneficial media additives. Although prior studies focused on byproducts that inhibit cell growth, this work sought to identify metabolites that can be added to improve process performance by increasing qP. Additives such as citrate offer the advantage of flexibility, in that they are easy to introduce at any stage, including during clone selection to provide a better indication of a clone’s potential performance. For example, when comparing clones B-2 and B-3 under the same platform process condition, differences in qP and volumetric titer appear to be negligible (Figure 4). However, with the addition of citrate, clone B-2 clearly outperforms B-3 with a 110% increase in final titer over the control condition without citrate. In comparison, other strategies for improving qP, such as genetic modifications or subcloning with higher levels of a selection agent require greater effort and time and have yielded only modest benefit to titer [28,53,54]. Furthermore, it is an open question if genetic modifications or a subcloning strategy effective for clones producing one mAb also apply to other clones producing a different mAb.

In conclusion, we demonstrate that high-qP associated extracellular metabolites identified by analyzing a library of multiple CHO cell clones producing different mAbs can be used to predict high-qP clones in another library. Through addback studies, we also show that a TCA cycle metabolite positively associated with qP can be used as a medium additive to improve both qP as well as final titer. To our knowledge, this is the first study to identify extracellular metabolites that specifically associate with qP across a large number of CHO cell clones and demonstrate their applications. The metabolomics approach presented here offers practical routes for identifying metabolites to improve clone selection and enhance overall productivity and titer.

## 4. Materials and Methods

### 4.1. Cell Lines

A total of 24 clones derived from a CHO GS KO parental cell line were used in these studies. The parental cell line has been described previously by Morris et al. [9]. Six clones expressing mAb A and six expressing mAb B were used for the initial fed-batch cell culture experiments to generate untargeted LC-MS data, and a subset of these 12 clones were used for addback experiments. Twelve other clones expressing mAb C were used for the targeted LC-MS experiment. All 24 clones were single-cell clones selected from multiple transfection pools to increase the likelihood of including diverse phenotypes with a wide range of growth and qP profiles.

### 4.2. Fed-Batch Cell Culture Experiments

Fed-batch experiments were performed on several platforms: 5 L bioreactors, 50 mL conical tubes, and 15 mL and 250 mL bioreactors (Ambr 15 and Ambr 250, Sartorius, Göttingen, Germany). The cells were grown using chemically defined BMS proprietary media. Temperature was maintained at 36.5 °C for all cultures. In bioreactors, dissolved oxygen was maintained above 40% using oxygen sparging and pH was maintained between 6.8–7.4 using sodium carbonate addition and carbon dioxide sparging. In addition to daily concentrated feed, boluses of glucose were supplemented starting on day 2 to maintain a concentration greater than 1 g/L. Cell suspension samples were taken daily over the course of 14 days to monitor growth using a Vi-Cell cell counter (Beckman Coulter, Brea, CA, USA). The daily samples were also analyzed for concentrations of lactate, ammonia, glutamate, and glutamine using a Cedex BioHT (Roche, Basel, Switzerland). Supernatants from the samples centrifuged at 1000× *g* for 5 min were analyzed by Protein-A HPLC for the antibody product.

### 4.3. Untargeted LC-MS

Supernatant samples collected on days 4 and 7 of the fed-batch experiments were diluted 1:10 with HPLC grade water and analyzed for metabolites using information-dependent acquisition (IDA) experiments performed on a quadrupole time-of flight (TOF) mass spectrometer (TripleTOF 5600+, AB Sciex, Framingham, MA, USA). Each sample was run four times with different combinations of chromatography methods and ionization modes (positive and negative). The chromatographic separation was performed with a binary pump HPLC system (1260 Infinity, Agilent, Santa Clara, CA, USA) using either a HILIC column (Phenomenex Luna NH2, Torrance, CA, USA) or a reverse-phase column (Phenomenex Synergi Hydro-RP, Torrance, CA, USA) as described previously [19]. The gradient methods and mobile phases are described in Appendix A. An example chromatogram can be found in Appendix A.

### 4.4. Feature Annotation

The LC-MS features were annotated with putative chemical identities based on accurate mass (*m*/*z*) and product ion (MS/MS) spectrum data. The details of the annotation procedure have been previously described [19]. Briefly, the *m*/*z* value and MS/MS spectrum of each unique feature was analyzed using the following five annotation tools: Metfrag [55], CFM-ID [56], NIST17 [57], Metlin [58], and HMDB [59]. For many features, these fives tools returned different annotations. To determine the most likely identity for a feature, we mapped each feature to one or more compounds in a model of CHO cell metabolism based on accurate mass and computed a score reflecting the confidence in the mapping. Some features were putatively identified as the same metabolite in multiple LC-MS methods. For example, tryptophan was detected in negative ionization mode paired with the Synergi column and positive mode with the HILIC column. In these cases, the putatively identified metabolite was represented by the feature responses (i.e., peak areas) from the LC-MS method that detected the highest dynamic range in peak areas across all samples.

### 4.5. Data Analysis

The peak area, representing the integrated area under the curve (AUC) of the extracted ion chromatogram, for each annotated feature was normalized to the integral viable cell density of the corresponding sample. These normalized AUCs were used in correlation analyses with peak VCD and qP calculated from early stationary phase (day 6 or 7, depending on experimental constraints) to mid-stationary phase (day 11 or 12). Before and after this period, respectively, the cells produce little antibody and show significant viability loss. All correlation analyses were performed only within experiments using data from the same timepoints. More detailed qP profiles are available in Appendix A. The qP was calculated using the following equation:$$\mathrm{Specific}\;\mathrm{productivity}\;qP\;\left[\frac{pg}{day\cdot cell}\right]=\frac{titer_{2}-titer_{1}\;}{iVCD_{2}-iVCD_{1}}$$

Other equations for *qP* were also tested, including a simple *titer*/*iVCD* calculation, and all provided similar results.

Both Pearson and Spearman correlation coefficients were calculated to test for linear and nonlinear relationships in the data, respectively. The *p*-values were controlled for false discovery rate (FDR) using the Benjamini–Hochberg method [60]. A *p*-value < 0.05 indicated a significantly correlated metabolite. Similar results were obtained using both correlations. For simplicity, the Pearson correlation coefficients are reported here. The same correlation analysis was also performed on the delta (difference) between day 7 measurements and day 4 measurements.

For pathway enrichment analysis, Metaboanalyst 4.0 was used, supported by the Mus musculus pathway library as the closest organism available [25]. Default selections were used for all other settings: the method for the over-representation analysis was the hypergeometric test, and the node importance measure for pathway topology analysis was relative betweenness centrality.

### 4.6. Targeted LC-MS

Multiple reaction monitoring (MRM) experiments were performed on a triple quadrupole mass spectrometer (6410, Agilent, Santa Clara, CA, USA) for targeted analysis of significantly correlating metabolites identified from the correlation analysis. High purity standards were used to optimize the following MRM parameters for each target analyte: ionization mode, precursor ion, fragmentor voltage for the precursor ion, product ion (i.e., MRM transition), and collision energy for the transition (Appendix A). For sample analysis, supernatants were diluted 1:10 and the same HILIC method was used as in the untargeted experiments. The samples were then analyzed in MRM experiments using the optimized acquisition parameters. Example chromatograms can be found in Appendix A.

### 4.7. Addback Experiments

To determine the effects of significantly correlating metabolites on specific growth and productivity, addback experiments were carried out in 50 mL conical tubes and the Ambr 250. The fed-batch cell culture experiments were performed as described above, except that varying doses of selected compounds were added in bolus to the culture medium on day 3. Each compound was added from a stock solution in a low, medium, or high concentration according to a D-optimal design (Appendix A). Control conditions used water addition to maintain the same volume as test conditions. Multiple concentrations were used to determine whether the cultures would exhibit a dose-dependent response in growth or protein production.

## Figures and Tables

**Figure 1 metabolites-11-00823-f001:**
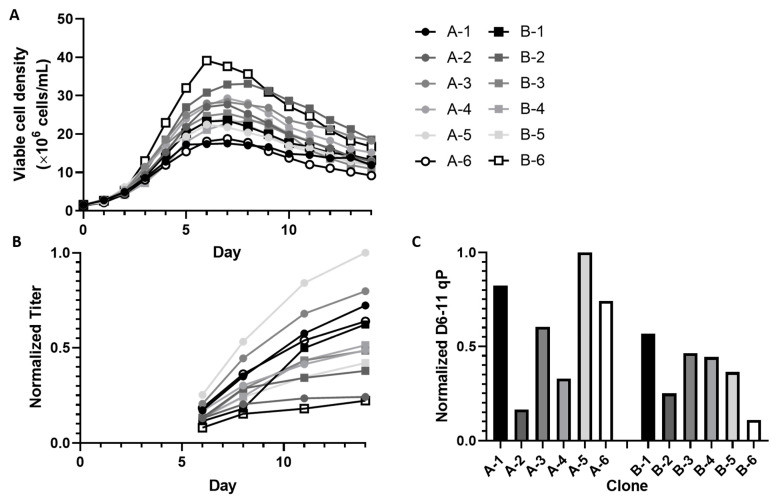
Growth and productivity profiles of 12 clones producing antibody A or B cultured in fed-batch bioreactors under identical process conditions. (**A**) Viable cell density (VCD) was recorded daily. (**B**) Titer was measured in samples collected on days 6, 8, 11, and 14. (**C**) Average qP from day 6 to 11 was calculated from integral VCD and titer.

**Figure 2 metabolites-11-00823-f002:**
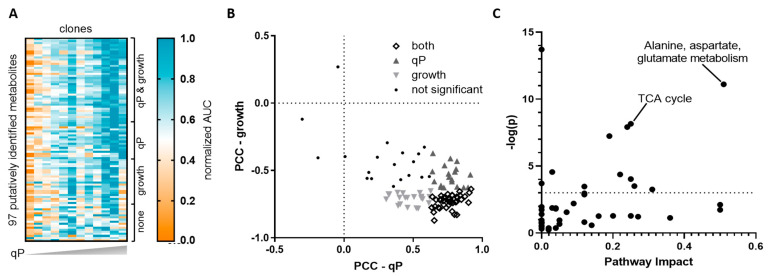
Significant correlations and pathways of extracellular metabolites from untargeted analysis. (**A**) Heatmap of integrated peak areas (AUCs). The AUC of a feature was first normalized to the corresponding sample’s viable cell density (VCD), then scaled to the maximum value for the feature across all samples. Clones are ordered from low qP to high qP, while rows are grouped from the top in the following order: correlated with both qP and growth, correlated with qP, correlated with growth, and no significant correlations. (**B**) Scatter plot of Pearson correlation coefficients (PCC) showing significance of correlation for each metabolite with peak VCD (*y*-axis) and qP (*x*-axis). Significant correlations with qP (▲), growth (▼), or both (**◊**) are indicated if the PCC has a *p*-value less than 0.05. Metabolites without any significant correlations are shown by filled dots (●). (**C**) Enrichment (*y*-axis) and pathway impact (*x*-axis) analyses using Metaboanalyst were performed on metabolites significantly correlated with qP, including those also correlated with growth. Labels indicate the top four ranked pathways as determined by pathway impact scores from betweenness centrality and significance in pathway enrichment (dotted line represents *p* < 0.05 calculated by modified Fisher’s exact test).

**Figure 3 metabolites-11-00823-f003:**
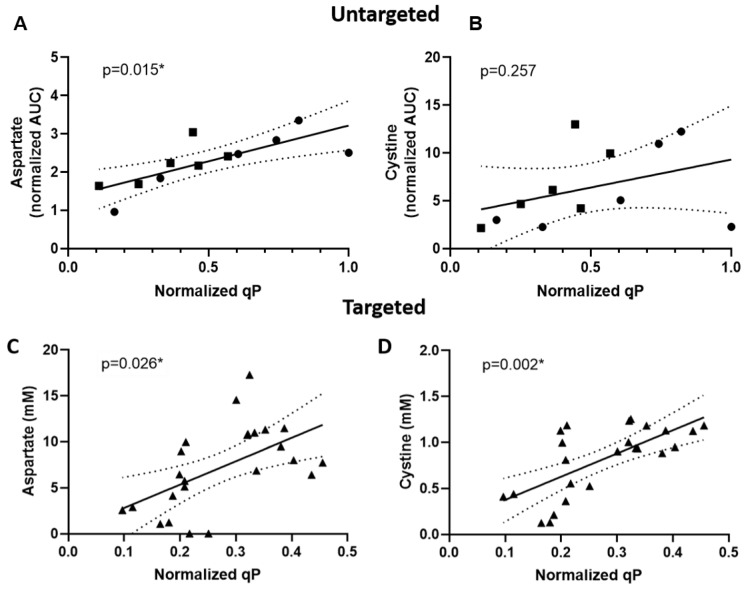
Correlations between day 7 metabolite levels and qP. Aspartate (left panels) and cystine (right panels) levels plotted against specific productivity (qP) for 12 clones expressing mAbs A (●) or B (■) cultured in 5 L bioreactors (**A**,**B**, individual bioreactors) or 12 clones expressing mAb C (▲) cultured in Ambr 250 bioreactors (**C**,**D**, duplicate bioreactors), with * indicating a *p*-value of <0.05. All qP values were normalized to the maximum qP among all 24 clones. For the untargeted experiment, metabolite levels shown are integrated areas under the curve (AUCs) from extracted ion chromatograms, normalized to VCD. Solid and dotted lines show, respectively, the ordinary least squares regression model and 95% confidence interval for the regression line.

**Figure 4 metabolites-11-00823-f004:**
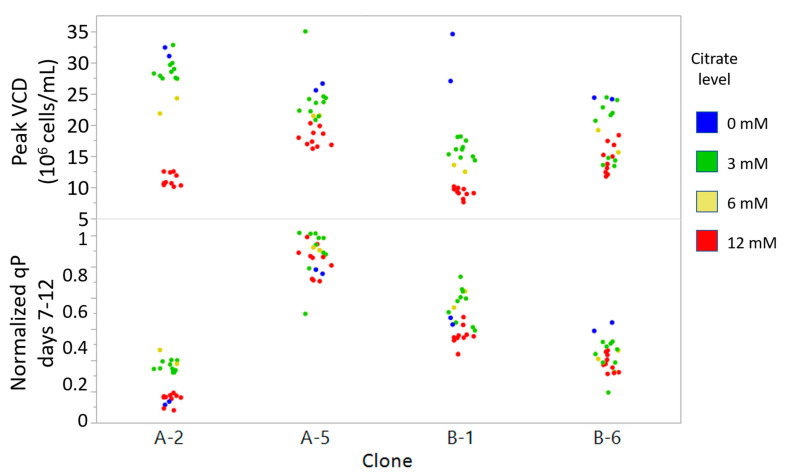
Effects of citrate addition on growth and productivity of selected clones with the highest (A-5, B-1) and lowest (A-2, B-6) qP for mAbs A and B. Citrate was added at 3 different levels as described in Appendix A. Grouping of conditions with the same level of citrate indicate dose-dependent changes in both peak VCD and qP.

**Figure 5 metabolites-11-00823-f005:**
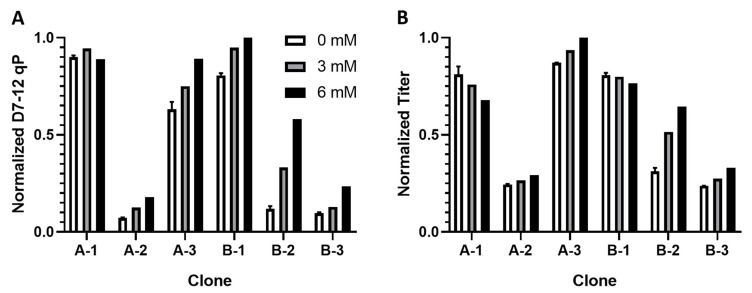
Responses of clones expressing mAb A (A-1 through A-3) or mAb B (B-1 through B-3) to addition of citrate to the culture medium. (**A**) Five out of six clones, each expressing one of two monoclonal antibodies (mAb A or mAb B, indicated by Clones A-1 through 3 and B-1 through 3), showed a dose-dependent increase in qP. (**B**) Four of the clones showed an overall increase in final volumetric titer. Data shown are normalized qP or titer scaled to the highest respective value across the six clones. Error bars show standard deviation across duplicate control cultures.

## Data Availability

Not applicable.

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
