# Peer review of "A Metabolomics Approach to Increasing Chinese Hamster Ovary (CHO) Cell Productivity"

_metabolites, 2021, doi:10.3390/metabo11120823_

Round 1

Reviewer 1 Report

This article presents an interesting application of metabolomics to cells in culture, to determine biomarkers of increased productivity and find potential routes to optimize bioreactor conditions. Unfortunately, the presentation of the methods, data and results is somewhat disorganized. The link between the three different part of the study (untargeted metabolomics, targeted analyses and medium supplementation) is not made clear enough and the article feels disjointed in this respect. It is not clear enough which metabolites were flagged for untargeted analyses and which were confirmed to have same trend using targeted analyses. 

The analysis of cystine: is this  function of the supplemented cystine in media or oxidized cysteine from cells? Why would the first and second analyses not correlate for this metabolite? How can you measure cysteine and cystine without protecting the cysteine in samples against oxidation?

Figure2: why do we see qP in bottom left corner of heat map? What does this heat map add to the paper? would be better to have a table in supplemental data showing real data instead of a heat map not showing anything specific as a result. It doesn't give any specific information about clusters of metabolites. 

The final confirmation of metabolites with reference standards is not explained explicitly enough. There should be a list of these metabolites, there matching RTs, and spectral match scores.

What are the 34 metabolites targeted in the second analysis? Can you show this data in a better way? The reader is left without much real proof to support the "results" shown in this paper.

Figure 4. Instead of citrate level 0 1 2 3, can you specify concentrations?

I suggest a revision done throughout the manuscript to reorganize the content and presentation of results to help clearly communicate the final conclusions that can be made from this work.

Supplemental methods need to be much more detailed (chromatography conditions, including columns used and mass spectrometry parameters. Representative chromatograms should be shown for untargeted and targeted analyses (in supplemental data)

Figure S6. how were these concentrations calculated? Is this well explained in the methods?

Reviewer 2 Report

In this manuscript the authors describe that, improvements in monoclonal antibody production has focused largely on increasing cell density within a reactor. Here, they used metabolomics to determine biomarkers, and thereby possible substrates, to increase yield per cell. Many identified metabolites are highly related to the TCA cycle and boosting with citric acid helps increasing yield.

The general findings of this manuscript are very interesting, however, the methods and interpretation of the data draw some comments regarding the authors work. These comments are stated below:

  1. Where there no internal standards used in the metabolomics measurements? How did the authors correct for extraction efficiency and analytical robustness of the measure metabolites?
  2. All the measurements have been performed on medium, why were the measurements not performed on the cells themselves? This is easily possible and would give direct insight into the metabolism.
  3. In line 20 the authors mention the following: “…again found that aspartate and cystine”. Cystine is not previously mentioned, though does have a direct link to alanine (cystine -> cysteine -> alanine).
  4. Line 85: Comment on the statement made here: It makes sense that both growth and production of antibodies costs energy, with an optimum somewhere in between having enough cells for production and investing enough energy into antibody production. Also: the next sentence is a bit confusing after saying “not always correlated”. If the highest growth has the lowest titer, it would make sense that there is a negative correlation between VCD and qP (each of the many cells made very little antibodies). So that would fit with a correlation between highest growth and titer, just a negative one. Right? Or is growth (rate?) and VCD not the same? This is a little bit confusing. Especially, since at line 138 they basically explain this idea. I would simplify.
  5. Line 104: Change “concentrations” to “abundance”. Concentrations are not determined here, as there are no calibration curves.
  6. Line 131: It’s probably helpful to indicate “higher metabolite levels in medium” as it might be easy to overlook that the levels in the cells may or may not behave in the opposite direction.
  7. Line 143: To name all the pathways is a bit misleading, when it’s mostly just overlap between targets. Succinate is a dicarboxylate, and in glyoxylate pathway, and TCA metabolite. Arginine metabolism, I would only include if the targets were directly related to it. If it’s fumarate and aspartate and glutamate -> TCA cycle.
  8. Line 143: Same as point 7, this all ties back into the TCA cycle.
  9. Line 195: This is not very clear. The low producing B is named, so that’s B-6 according to figure 4. Low dose, I assume is then the lowest dose (specify); green dots. “improving qP two-fold”. But the green dots are below the blue dots? Minimal impact on growth also seems weird, as quite a few green dots are at the very bottom of VCD levels. Please clarify; also use sample codes in text for ease of reading.
  10. Line 263: If the cause is accumulation of toxic products, rather than depletion, then why would supplementation of precursors alleviate the issue?? (citrate -> succinate -> fumarate -> malate -> oxaloacetate -> aspartate)
  11. Line 321: The author are not making big claims here and seem to be aware of the implied paradox, but the sentence “support higher productivity” feels a bit misleading. The cited article is about adding cysteine to the feed. This part of the discussion obfuscates the difference between “end-point” concentrations and feed concentrations. This is more important for sulfur containing compounds than things like malate. Sulfur isn’t as readily available as carbon. The idea here is that because it’s increased in the medium, it’s increased in the cells themselves. But if cells were consuming it, levels would go down. So, what would then be the source? Protein in the medium? Are other sulfur containing amino acids going down? Was free cysteine measured as well and could it be a shift to cystine over cysteine? (if all conditions are equal between samples, the authors might still determine differences in cysteine, even though there is some loss to cystine dimerization). It could also be that less of it is used by the cells, but that makes the opposite point of what they are getting at with the references; that the more successful cells require fewer cystine, not more. It would seem unlikely that the cells get such a craving for cystine that they are making it and then are expelling half of it. I fully support basically all the info here, including the last sentence, but I would really change the wording to reflect this somewhat surprising finding.
  12. Line 398: pH was maintained at around 7. Are there records of how much carbonate was used for each reactor to keep pH stable? Did this have any correlation with the expelled biogenic acids or cystine?
  13. Figure 2: Figures A and B do not show metabolite names, so it’s merely to show that A: there are differences between strains, and B: some differences are significant? This is not made clear from these figures.
  14. Figure 4: why does this say “citrate level 0 1 2 3” and figure 5 then gives the amounts in mM? Also, why isn’t the format between 4 and 5 the same?

Reviewer 3 Report

The manuscript titled "A metabolomics approach to increasing CHO cell productivity" by Yao and collaborators presents a very interesting strategy to identify the nutrients and metabolites which influence the volumetric and specific productivity of monoclonal antibodies in CHO cell cultures.

Initially, untargeted metabolic profiling was performed on a panel of 12 clones producing one of two mAbs. This study identified 97 unique CHO cell metabolites and enabled their relative quantification at two culture phases, mid-exponential and early stationary growth. The quantified metabolites were then subjected to correlation analyses. The authors report 79 metabolites that correlate with cell specific productivity (qP), cell growth or both. Of these 79 metabolites, only a quarter is present in the basal or feed medium and, therefore, presumably act as nutrients.

Ambr15 microbioreactor cultures of a second panel of 12 clones producing a third mAb were conducted to confirm whether the metabolites identified in the initial, untargeted study could be used to identify high-qP clones. In this study, 34 metabolites were observed to positively correlate with peak VCD and/or qP. Of these, the authors highlight aspartate and cysteine as having a strong positive correlation with qP and, therefore, as being potential indicators of clones with high qP.

Based on the above results, the authors performed a third round of CHO cell cultures, where four cell lines were cultured with varying concentrations of aspartate, aminobutyrate, citrate, and glutamate to yield a total of 96 conditions. The only metabolite observed to have a strong positive correlation with qP was citrate, although it was also found to negatively impact growth.

The final experiment involved culturing three clones expressing mAb A and three expressing mAb B fed with a narrower range of citrate concentrations in Ambr250 bioreactors. Results from this study confirmed that increased citrate feeding enhances mAb qP in five of the six clones and yields higher mAb titre in four of the six clones. The authors also report that, when citrate increases qP, a concomitant accumulation of Gln, Glu, and Lac and a reduction in ammonia production is observed.

The design, extent, and quality of the experimental work is excellent, and data analysis and discussion are commensurate, quality-wise, with the experimental work. Overall, I believe the manuscript presents a robust strategy to identify clones that yield enhanced mAb (specific and volumetric) productivity. In addition, the manuscript also identifies citrate as a metabolite which, when fed, can increase mAb qP.

Before recommending publication of the manuscript, I am interested in knowing why the authors opted for performing the correlation analyses using iVCD-normalised AUC instead of the cell-specific AUC rates. I ask this question because, as the authors indicate in Section 3.2 (lines 274-276), metabolite accumulation can arise from reduced uptake by the cells. In this context, it may be that metabolites whose AUC correlates positively with qP (especially those which are fed to culture) do so because their uptake rate is inversely proportional to qP. It could be argued that correlation analysis using specific rates (ΔAUC/ΔiVCD) may yield further information on the bottlenecks of mAb qP.

I acknowledge that iVCD would be required to compute the specific rates and could, thereby, lead to increased co-dependency between the compared data. However, the authors already normalise AUC to iVCD prior to correlation analysis, so using iVCD to compute specific uptake rates may not cause issues. I also acknowledge that there may not be sufficient metabolite datapoints to calculate specific rates during mid-exponential growth. However, specific rates could be computed for the interval between mid-exponential and early stationary phases.

Along a similar vein: why use peak cell density as a proxy for growth instead of iVCD?

I believe it would be valuable if the manuscript had a brief comment on why iVCD-normalised AUC was used for correlation analysis instead of cell-specific AUC rates. Once this discussion is included, I believe the manuscript will be ready for publication in Metabolites, will be of high interest for the journal’s readership, and would be an important resource for the CHO cell culture community.

Round 2

Reviewer 1 Report

Here are some items that would need revising:

1) a few listed metabolites do not make sense as metabolites in cells (highlighted in supplemental file with comments), such as capecitabine, dhurrin, hydroxymandelonitrile, fluoromucunolactone, GSH episulfonium ion. The authors should discuss the presence of these metabolites if they want to keep them in the list. 

2) Add table of list of "identified" metabolites with m/z, ppm, RT, library scores.  Clearly state how many "identified" metabolites from untargeted analysis were verified with synthetic stds.

3) Also, add more method information on how metabolites were identified from untargeted data and which were confirmed by synthetic standards

4) LC methods: add the information on which column was used for both separations

5) MS method: source conditions missing

Author Response

Here are some items that would need revising:

1) a few listed metabolites do not make sense as metabolites in cells (highlighted in supplemental file with comments), such as capecitabine, dhurrin, hydroxymandelonitrile, fluoromucunolactone, GSH episulfonium ion. The authors should discuss the presence of these metabolites if they want to keep them in the list. 

We agree with the reviewer that these are unlikely endogenous to CHO cells. They were putatively annotated by BioCAn because they were catalogued in KEGG as metabolites able to be produced by CHO enzymes, but we carefully reviewed the list and removed non-biological metabolites including these 5 and deoxy-fluorocytidine. In the main text we revised the total from 79 significantly correlated metabolites to 73 to account for this.

2) Add table of list of "identified" metabolites with m/z, ppm, RT, library scores.  Clearly state how many "identified" metabolites from untargeted analysis were verified with synthetic stds.

We added Table S4 of the identified metabolites including a column indicating standard confirmation.

3) Also, add more method information on how metabolites were identified from untargeted data and which were confirmed by synthetic standards

These metabolites were annotated using BioCAn as described elsewhere (reference 19), which is described in the methods section. To be clear, many of these start as only putative identities, and that is why we call them “annotations” or “putatively identified metabolites” in the text. Of these, 22 reached level 1 identification through confirmation with synthetic standards as seen in Table S4 referenced in 2).

4) LC methods: add the information on which column was used for both separations

We have added the column names to the Supplementary Information.

5) MS method: source conditions missing

We have added ion source conditions to the Supplementary Information